# Effective Carbon Dioxide Mitigation and Improvement of Compost Nutrients with the Use of Composts’ Biochar

**DOI:** 10.3390/ma17030563

**Published:** 2024-01-25

**Authors:** Sylwia Stegenta-Dąbrowska, Ewa Syguła, Magdalena Bednik, Joanna Rosik

**Affiliations:** 1Department of Applied Bioeconomy, Wrocław University of Environmental and Life Sciences, Chełmońskiego Street 37a, 51-630 Wrocław, Poland; sylwia.stegenta-dabrowska@upwr.edu.pl (S.S.-D.); ewa.sygula@upwr.edu.pl (E.S.); 2Institute of Soil Science, Plant Nutrition and Environmental Protection, Wrocław University of Environmental and Life Sciences, Grunwaldzka Street 53, 50-375 Wrocław, Poland; magdalena.bednik@upwr.edu.pl

**Keywords:** composting, biochar, GHG emissions, process intensification, carbon monoxide

## Abstract

Composting is a process that emits environmentally harmful gases: CO_2_, CO, H_2_S, and NH_3_, negatively affecting the quality of mature compost. The addition of biochar to the compost can significantly reduce emissions. For effective CO_2_ removal, high doses of biochar (up to 20%) are often recommended. Nevertheless, as the production efficiency of biochar is low—up to 90% mass loss—there is a need for research into the effectiveness of lower doses. In this study, laboratory experiments were conducted to observe the gaseous emissions during the first 10 days of composting with biochars obtained from mature composts. Biochars were produced at 550, 600, and 650 °C, and tested with different doses of 0, 3, 6, 9, 12, and 15% per dry matter (d.m.) in composting mixtures, at three incubation temperatures (50, 60, and 70 °C). CO_2_, CO, H_2_S, and NH_3_ emissions were measured daily. The results showed that the biochars effectively mitigate CO_2_ emissions during the intensive phase of composting. Even 3–6% d.m. of compost biochars can reduce up to 50% of the total measured gas emissions (the best treatment was B650 at 60 °C) and significantly increase the content of macronutrients. This study confirmed that even low doses of compost biochars have the potential for enhancing the composting process and improving the quality of the material quality.

## 1. Introduction

Industrial composting is one of the most environmentally friendly ways to manage green waste. Due to the biological nature of the composting process, it leads to the emission of harmful gaseous compounds such as greenhouse gases (GHGs). During composting, oxygen (O_2_) is consumed and carbon dioxide (CO_2_) is released, as well as other volatile substances produced by microorganisms such as methane (CH_4_) and nitrogen oxide (N_2_O) [1]. CO_2_ is the main gas generated during the microbial decomposition of organic matter, which plays an essential role in the balance of heat, and the rise in atmospheric CO_2_ levels will have an impact on climate change [2]. CH_4_ is produced under anaerobic conditions by methanogens. It has 26 times the capacity of CO_2_ to absorb infrared light and exacerbate the greenhouse effect, making it a powerful greenhouse gas [3]. The composting process also generates ammonia (NH_3_), which is a critical precursor to atmospheric particulate matter [4]. In the nitrogen conversion process, it is unavoidable to produce N_2_O and NH_3_. N_2_O is a significant contributor to the depletion of the ozone layer. The warming potential of a single molecule of N_2_O is 296 times higher than that of CO_2_ [5]. Greenhouse gas emissions from composting are not only harmful to the environment but also negatively affect the final parameters of the stabilized material [6]. Several studies have revealed that organic waste from pesticide use contains a significant amount of mobile heavy metal fractions. This poses a risk of soil contamination and limits the possibility of their cyclic use since heavy metals are densified during the composting process [7].

In recent years, biochar has gained popularity as an effective tool for reducing gaseous emissions due to its special physicochemical properties. Biochar is a carbon-rich product from biomass burning in the absence of oxygen, called pyrolysis. This pyrolysis product is gaining attention due to its polyaromatic and microporous structures, large specific surface area, varied surface functional groups, and high cation exchange capacity, as described in the recent literature [8,9,10,11]. Biochar is widely recognized as a promising material for improving soil fertility and sequestering carbon on the centenary time scale, thereby reducing climate change [12,13,14]. In addition, biochar increases aggregation and the soil’s ability to retain water, which is particularly positive for plants growing under water stress, increasing grain yield [15]. In the case of composting, biochar can be a particularly useful tool that positively influences the course of the aerobic stabilization process [16]. The application of biochar has a significant impact on the moisture content, oxygen availability, temperature, pH, and C/N ratio during the composting of organic waste. Additionally, biochar promotes the aeration rate, enhances gas exchange, and prevents the formation of anaerobic zones, which lead to a significant reduction in greenhouse gases and odor emissions during the composting process [17]. Biochar has a high potential for CO_2_ mitigation, is durable, and has low reversal risk. It may also benefit food security and soil fertility when applied to soil [18,19]. In addition, the remarkable properties of biochar make it a particularly useful tool for the sorption of heavy metals, which reduces soil ecotoxicity [20,21,22].

So far, the most popular substrates for biochar production have been waste biomass with a high proportion of organic matter. However, the amounts of biochar added to compost to effectively remove CO_2_ could even be 19% [23], while the production efficiency of such wood biochars is extremely low—up to 90% weight loss during production [24]. This has made the addition of such biochars less effective. There is still a lack of studies outlining the use of biochars made from biomass with a higher proportion of the mineral fraction-like compost and their effect on the composting process. The first studies are emerging on the promising effect of home compost biochars on the adsorption of volatile organic compounds (VOCs) emitted during food waste storage. The compost used in this study was primarily characterized by a low organic content (<30%), which caused a change in the specific surface area of the biochar and improved the adsorption process capabilities [25]. 

This study aims for the first time to test the compost biochars’ capabilities to reduce greenhouse gas emissions (GHG), including CO_2_ mitigation, and to improve the quality of compost by increasing the concentration of nutrients in the composting matrix. The authors provide recent findings on the GHG mitigation from the composting process by composts’ biochar. Furthermore, this study provides recommendations for use of compost biochars for lowering the specific gaseous compounds emitted during the intensive phase of composting. Overwhelming evidence and benefits of using compost biochars are presented to improve the composting process and its ability to retain heavy metal contaminants.

## 2. Materials and Methods

### 2.1. Feedstock and Biochar Characteristics

The compost biochars were prepared using a laboratory muffle furnace (SNOL, model 8.1/1100, Utena, Lithuania) in retention time 1 h, temperature 550, 600, and 650 °C, heating rate of 10 °C∙min^−1^; biochars were labeled B550, B600, and B650. The material used for compost biochars’ production was mature, certified compost originating from a composting plant (Best-Eko; Rybnik, Poland). CO_2_ was supplied into the chamber during the entire pyrolysis process to maintain an inert atmosphere. After carbonization, the furnace was turned off and left to cool. For this study, the compost biochars’ samples were sifted through a 1 mm sieve. As a feedstock for the composting process in the experiment, a mixture of green waste and sewage sludge in a ratio of 9:1 was used originating from the composting company (Best-Eko, Rybnik, Poland).

For feedstock and biochar characteristics, some basic analyses were performed: pH in water solution in 1:10 mass ratio (Elmetron, CPC-411, Zabrze, Poland) and moisture content (MC) according to PN-EN 14346:2011 (laboratory dryer, WAMED, model KBC-65W, Warszawa, Poland) [26,27]. Ultimate analysis of elemental composition C, H, N, S was conducted according to PN-EN ISO 16948:2015-07 (Perkin Elmer, model 2400 Series, Waltham, MA, USA) [28]. The O content of compost biochars and feedstocks was determined by mass balance (O% = 100 − C% − H% − N% − ash%) [29,30].

Near-total concentrations of macronutrients (Ca, Na, Mg, K, P) and selected metals (Zn, Cd, Pb, Ni, Cr, Cu, Mn, Hg) were determined after digestion with aqua regia. Briefly, 1 g of the sample was placed in a glass falcon and 2 mL of 30% hydrogen peroxide was added, to remove the excess of organic matter. Then, samples were treated with 7.5 mL of concentrated HCl and 2.5 mL of concentrated HNO_3_, left overnight and boiled for 5–6 h [31]. After the procedure has finished, the obtained suspension was filtered and filled up with distilled water to the volume of 50 mL. In the extracts, the content of Ca, Na, Mg, and K was determined on MP-AES 4200 Spectrometer (Agilent Technologies, Santa Clara, CA, USA) [32]. The other 8 elements: Zn, Pb, Cd, Cu, Cr, Ni, Mn, and P were examined on an ICP-OES (iCAP 7400, Thermo Scientific, Waltham, MA, USA) (PN-EN ISO 11885:2009 [33]). A separate procedure was applied for Hg, whose concentration was determined in the solid sample, using atomic absorption spectrometry method with the amalgamation technique (MA-2 analyzer; Nippon Instruments Corporation, Osaka, Japan).

For compost biochars’ samples, the specific surface area (SSA) and FTIR-ATR spectroscopy was also measured. SSA were performed with sorption of N_2_ in 77K (Micromeritics ASAP 2020; Norcross, GA, USA). Augmented total reflection–Fourier-transform infrared (ATR)-FTIR measurements were performed with a Nicolet iN10 integrated infrared microscope with Nicolet iZ10 external FT-IR module (Thermo Fischer Scientific, Waltham, MA, USA) equipped with a deuterated-triglycine sulfate (DTGS) detector and a diamond ATR module. For each spectrum, 32 scans were averaged in the mid IR range of 400–4000 cm^−1^ at a spectral resolution of 4 cm^−1^.

### 2.2. Experiments Configuration

The 15 g (±0.5 g) mixture of feedstock (green waste and sewage sludge in a ratio of 9:1), was placed in the 1 dm^3^ reactors (Figure 1), with the intended dose of compost biochars to individual reactors. Then, the reactors were placed in a thermostatic cabinet for incubation at the following temperatures: 50 °C, 60 °C, 70 °C (Figure 1). Gas concentration measurements were made using a portable gas analyzer in each 24 h. The measured concentrations were later converted into emissions according to the equations in Section 2.3. Biochars were added to the reactors in the following doses: 0%, 3%, 6%, 9%, 12%, and 15% per weight on a dry basis (Figure 1). The compost biochars made from compost were obtained at different temperatures and different rates of temperature increase: 550 °C 10 °C∙min^−1^, 600 °C 10 °C∙min^−1^, 650 °C 10 °C∙min^−1^, with the 1 h retention time. The decisive factor in choosing these variants was their specific surface area (research on the selection of specific types of biochars for testing was not an element of this engineering work). At the same time, during gas emission tests in the reactors, the larger sample of substrate was incubated in the same conditions in the amount of approximately 350 g ± 5 g, which was used to perform post-process analyses—pH, loss of ignition (LOI), moisture content (MC), ultimate analysis CHNS, elemental analysis: Zn, Cd, Pb, Ni, Cr, Cu, K, P, Mn, Ca, Mg, Na, and specific surface area (SSA) (only for compost biochars’ samples).

### 2.3. Analyses of Process Gas Emissions

Measurements of selected gases (CO_2_, CO, H_2_S and NH_3_) concentrations during the aerobic digestion process were carried out for 10 days, once a day, using a portable electrochemical gas analyzer (Nanosens DP-28 BIO; Wysogotowo, Poland). Concentrations of CO, H_2_S, and NH_3_ were determined in ppm in the following ranges: CO 0–2000 ppm (±20 ppm), H_2_S, NH_3_ 0–1000 ppm (±10 ppm), and CO_2_ 0–100% (±2%). The measurements were made on a reactor previously removed from the thermostatic cabinet. Each measurement lasted 45 s, followed by automatic cleaning of the analyzer. After the measurements, the cover of the reactor was opened for 30 min to aerate the material. The following formulas were used to convert measured concentration of gases for emissions:Conversion to volume, for H_2_S, CO and NH_3_ (ppm):
(1)$$V=\frac{1000\times M\times (1.66\times 10^{-24})\times ((2.68839\times 10^{22})\times a)}{1,000,000}$$
where
V—gas volume;M—molar mas;a—accumulated concentration, ppm.
Conversion to volume, applies to CO_2_ (%) only:
(2)$$V=\frac{1000\times M\times (1.66\times 10^{-24})\times ((2.68839\times 10^{22})\times a)}{100}$$
where
V—gas volume;M—molar mas;a—accumulated concentration, ppm.
Specific emissions, applies to H_2_S, CO, NH_3_ (ppm):
(3)$$E=\frac{V}{DTS\times 1000}$$
where
E—emissions, µg∙g d.m.;V—gas volume;DTS—dry total solids.
Specific emissions, applies to CO_2_ (%) only:
(4)$$E=\frac{V}{DTS}$$
where
E—emissions, µg∙g d.m.;V—gas volume;DTS—dry total solids.

### 2.4. Statistical Analysis

For statistical analysis, 13.3 Statistica software (TIBCO Software Inc., Palo Alto, CA, USA) was used. For the statistical differences of the contribution of VOCs, one-way ANOVA was applied according to Tukey’s test at a significance level *p* < 0.05, including previous verification of normality and homogeneous variance using the Levene test. For all relevant cases, standard deviation (SD) was applied.

## 3. Results and Discussion

### 3.1. Compost, Feedstock and Compost Biochars’ Characteristics

For biochar production, commercial certified compost from the Best-Eko composting company (Rybnik, Poland) was used, as a potential source of nutrients and a good material for gas emission mitigation, with beneficial properties enhanced in the pyrolysis process. Used compost has a typical pH value—near to neutral (Table 1)—and it is observed that with the temperature of pyrolysis, the pH increased. Biochars have a strongly alkaline pH, from 8.61 in variant B550 to 9.50 in B650. The increase in pH with the temperature of pyrolysis is typical, and similar results were observed with the pyrolysis of Japanese larch (*Larix kaempferi Sarg*.) and dairy manure (up to 11 pH) [34]. The pH values of biochars are positively correlated with the formation of carbonates and the contents of inorganic alkalis. These groups are the main cause of the alkaline reaction [35]. Higher pH has been associated with the increase in ash content and formation of oxygen functional groups that occur during the pyrolysis process. The feedstock (mix of green waste and sewage sludge) has an acidic reaction with pH 4.77, which is not optimal in composting, but is mostly observed for green waste [36]. This indicates a potentially promising use of the addition of compost biochars in neutralizing the acidic reaction of the substrate.

The compost was characterized with a huge content of mineral fraction (almost 75%) and this content increased in biochars with the pyrolysis temperature. Biochars derived from livestock manure also have more inorganic fraction—e.g., chicken manure, 40.6% d.m. [37]. In other studies, mixes of chicken manure, clay, minerals, and specific organic compounds were used to produce the organo–mineral biochar [38]. This type of biochar studied by other authors was shown to increase soil mineral content, improve soil cation exchange capacity (CEC), decrease the mineralization rate of organic matter, as well as having potential to maintain nutrients longer in the soil compared to a pure biochar.

The compost used in this study was rich in macronutrients: Ca (17,760 mg∙kg^–1^), K (6155 mg∙kg^–1^), Mg (2697 mg∙kg^–1^), Na (1793 mg∙kg^–1^), and P (5835 mg∙kg^–1^), which is typical for high inorganic content biochars [37]. In produced compost biochars, the content of nutrients become concentrated and follow the order: B650 < B600 < B550. The same trend was observed for metal content—all of them increased, except the Zn, which decreased in biochars below the level measured in compost. The content of C, H, N, S, and O decreased with the pyrolysis process, which is typical for biochars produced from substances with a low organic content and was previously observed, for example, in sewage sludge [39]. More importantly, the produced biochars were rich in P, which is a crucial element for living organisms, and there is a decreasing supply of phosphate on the market [40]. Due to the high content of P in biochars, future compost is potentially valuable for use as a fertilizer. At the same time, the metal content in all analyzed materials was significantly below the standard for fertilizers in Poland and in the EU regulations [41].

The important parameter in the assessment of biochars’ absorption properties is their specific surface area (SSA). In the present study, values of SSA (Figure 2) sharply increased along with the temperature of pyrolysis, from 6.06 (B550) to 39.2 m^2^∙g^–1^ (B650). Kumar et al. [16] showed that, in general, municipal solid biochars and livestock manure biochars have a lower SSA and porosity than plant biochars, e.g., the authors of [42] observed an increase in SSA parameters for bamboo biochars to 181.05 m^2^∙g^–1^ at 600 °C, in comparison to chicken manure biochars—11.84 m^2^∙g^–1^ at 500 °C [37].

#### FTIR-ATR Spectroscopy

The spectra of the compost sample shows (Figure 3): a band at 3320 cm^–1^ that can be ascribed to C–H bonds and to O–H of alcohols, phenols, or O–H carboxyl and also to N–H vibrations in amide functions; two characteristic bands at 2925 and 2845 cm^–1^ that may be attributed to asymmetric and symmetric vibrations of C–H stretching of CH_3_ and CH_2_ groups [43]. The peaks at 1700 and 1600 cm^−1^ were assigned to C=O stretching for carboxyl groups and aromatic C=C bonds, respectively [44]. The peak at 1270 cm^−1^ was attributed to phenolic–OH stretching [45]. Typical for the pyrolysis, all of the bands listed above that were characteristic of compost disappeared, when no new bands typical of biochar samples appeared, as was previously observed in wooden biochars [34]. This could be an effect of changes in the proportion between mineral and organic fraction—at the end of pyrolysis, 90% of mineral fraction was observed. This indicates that some of the chemical bonds were broken. In compost biochars, when compared to raw compost, the peak at 1270 cm^−1^ attributed to phenolic–OH stretching was invisible, when strong absorption peaks at 750–870 cm^−1^ belonging to stretching vibrations of aromatic ring C–H or C–N, R–O–C, or R–O–CH_3_ groups appear [37]. The strong peak at around 1020 cm^–1^ indicates the combination of C–O stretching of polysaccharides, in addition to Si–O–Si bonds of silica, and to the group Si–O–C that was visible in all samples [44].

### 3.2. Effect of Biochars on Composting

#### 3.2.1. Effect of Biochars on pH and LOI during the Initial Stage of Composting

The material incubated for 10 days was subjected to pH and loss on ignition (LOI) testing (Appendix A). In the case of pH, values of a mildly alkaline reaction were observed, especially in the reactors incubated at 50 °C. In the case of biochar doses, an increase in the amount of hydrogen ions relative to the control sample was observed regardless of the production temperature. In the material incubated at 60 °C, it was observed that the pH values for the biochar produced at 650 °C differed significantly from the biochar produced at 550 °C and 600 °C. In the material with the addition of B650, a slightly more acidic environment was present than in the case of the other biochar; only for the dose of 15% was there an increase in pH, which finally amounted to 8.54. In contrast, for the materials incubated at 70 °C, the pH decreased, relative to the other temperatures, which was most likely due to the intensification of the composting process [46]. No significant correlations were observed between the biochar dose and the amount of hydrogen ions. The LOI values in the vast majority of cases ranged from about 70% d.m. to as high as 87% d.m., and a decrease in the value of the parameter was observed with the increasing biochar dose. In addition, this study showed that the initial LOI for biochar produced at 650 °C tended to be higher than for other production temperatures. High parameter values indicate a high content of organic matter in the material.

Sundberg et al. studied the effect of pH and microbial content on the odors of composted food waste. The researchers examined pH values for materials composted on days 3, 8, and 16 of the process, and incubated the material at 50 °C and 70 °C. In each case, the results indicated an increase in pH over time, while samples incubated at 70 °C reached higher values much faster [47]. On the other hand, Wasaq et al. optimized the composting of food waste using biochar produced at temperatures of 350 °C and 450 °C. Doses of 10% and 15% biochar were applied to the compost. The study showed that both the dosage and the type of biochar used had a significant effect on the final pH of the material after the process. The hydrogen ion content of the 10% dose was lower than that of the 15% dose, and a similar relationship was observed for the temperature of biochar production. The addition of biochar had a beneficial effect on the ammonification and denitrification process, which also correlated with an increase in the pH value of the composted materials [48]. Czekała et al., in their study on the co-composting of pig manure with the addition of biochar, also observed an increase in pH depending on the biochar dose [49]. Jia et al. [50], Jindo et al. [51] and Zhang [52] linked a rapid increase in pH to accelerated fatty acid degradation and NH_3_ emissions. The high pH value in the compost can also be linked to the high content of hydrogen cations in the biochar studied [44]. 

#### 3.2.2. The Emission Change

During the initial phase of composting, some potentially harmful gases are released. In this study, the CO_2_, H_2_S, NH_3_, and CO emissions, and the effect of different type of compost biochars and the doses were measured (Appendix A, Figure 4). During the thermophilic phase, the composting temperature can increase from 50 to 80 °C. The thermophilic phase T > 55 °C usually lasts 7–10 days, so 50, 60, and 70 °C temperature variants were tested to simulate the intensive phase of composting [53]. The observed emissions at variant 0—without the addition of biochar was:CO_2_
○429.1–565.0 mg∙g^−1^ d.m. for incubation at 50 °C;○410.8–522.5 mg∙g^−1^ d.m. for incubation at 60 °C;○153.4–270.9 mg∙g^−1^ d.m. for incubation at 70 °C.CO
○477.7–653.5 µg∙g^−1^ d.m. for incubation at 50 °C;○503.5–861.9 µg∙g^−1^ d.m. for incubation at 60 °C;○508.8–784.1 µg∙g^−1^ d.m. for incubation at 70°C.H_2_S
○60.4–85.5 µg∙g^−1^ d.m. for incubation at 50 °C;○129.4–191.5 µg∙g^−1^ d.m. for incubation at 60 °C;○52.8–85.3 µg∙g^−1^ d.m. for incubation at 70 °C.NH_3_
○0.0–34.1 µg∙g^−1^ d.m. for incubation at 50 °C;○59.0–188.5 µg∙g^−1^ d.m. for incubation at 60 °C;○0.0–119.1 µg∙g^−1^ d.m. for incubation at 70 °C.


##### CO_2_ Emissions

CO_2_ was the main gas generated during the bio-oxidative phase of composting. Additionally, CO_2_ is rarely produced as a pure component [54]. The adsorption of CO_2_ from gas mixtures requires knowledge of the impact of the other gases on the adsorption capacity. Gases can compete for adsorbent sites, form complexes with adsorbed compounds, and interact in the gas phase which will impact adsorption efficiency [55]. There are multiple articles available discussing biochar adsorption of pure CO_2_ from gas mixtures [55,56], although the use of biochar for CO_2_ capture in a composting matrix is described less [57]. The observed trend for CO_2_ emissions was similar at 50 and 60 °C, and dynamic from the beginning of the observation (Appendix A). In contrast, at 70 °C, the CO_2_ emissions became more static. The CO_2_ cumulative emissions were ~500 CO_2_ mg∙g d.m. at 50 and 60 °C, and <200 CO_2_ mg∙g d.m. at 70 °C (Appendix A). In other observations, after 10 days of composting, those emissions were similar (120–170 mg∙g d.m.), but the authors showed that the form of the applied biochars is important, especially after 10 days of composting—granulation of biochars result in CO_2_ mitigation [1]. On the other hand, the granular biochar resulted in the faster cooling of the composted material, which could mean that the smaller amount of CO_2_ was simply due to the slower decomposition process of the organic matter. In other research, the authors proved that the addition of bamboo biochar is effective in reducing CO_2_, but during the cooling and maturity period of the composting. Yan et al. proved that biochar can reduce the abundance of key CO_2_ emitting enzymes in the TCA cycle at the later stage of composting, thereby reducing carbon losses [58]. The addition of biochar usually decreased CO_2_ emissions especially in doses of 3 and 6% d.m. (Appendix A). The most effective treatment for the mitigation of CO_2_ emissions was the addition of biochars at a temperature of 70 °C, when the total reduction was above 80% compared to variants zero (without the addition of biochar). A much lower reduction was observed at 50 and 70 °C, usually <20%. At those temperatures, the most effective biochar was B650. Those observed reductions in CO_2_ were much higher than previously observed by Czekała et al. [49], which was 6.9 and 7.4% with a dose of 5 and 10%, respectively. In other research, emitted CO_2_ was 2 and 3% higher with the addition of biochars than in the control [59]. In other experiments, bamboo biochar amendments reduced CO_2_ emission by 16.77% [58].

Unfortunately, in some cases, increases in CO_2_ emissions were observed, especially at 70 °C, and with biochar doses above 10% d.m. (Appendix A). The most effective CO_2_ reduction with the addition of biochar was at a temperature of 70 °C—the total reductions were up to 80% (Figure 3). Ottani et al. observed that the addition of 3–5% d.m. biomass biochar emitted 5% more CO_2_ than the control, with the *p*-values of the Tukey test being not significant. Ottani et al. explained that the addition of biochars facilitated the aerobic degradation processes of the organic substance, causing an increase in CO_2_ in the first 10 days of composting [60].

##### CO Emissions

It was observed that emissions of CO increased with time and temperature (Appendix A), which was previously indicated with our own study [61]. This is an effect of the thermochemical reactions and biological activities of microorganism. The highest emissions of CO were registered at the beginning of the composting process, as found in other composting matrices like green waste, manure, and municipal solid waste [62,63]. The biggest emissions of CO were observed in variants with the addition of B650 ~2000 µg∙g d.m. All of the tested biochars show the highest emissions of CO at 70 °C (up to 1200 µg∙g^−1^ d.m.), in comparison to other variants where those cumulative emissions were ~500–800 µg∙g^−1^ d.m.

The best reduction in CO emissions was observed at a temperature of 50 °C, with a 30–60% reduction for B550, and 20–30% for B600. Variant B650 showed no clear effect, and this biochar was less effective, especially at 70 °C, when massive emissions of CO occurred (>300% increase in observed emissions compared to variant 0). Increasing the dose of B550 and B600 usually did not have a specific effect for the decrease in CO, only a slight reduction was observed with dose 3, and 12% on dry matter. B550 was the most effective in CO mitigation at all incubation temperatures. Although CO production through composting is well documented, observations of the addition of biochars for CO emissions is very limited. In this study, significant changes in CO emissions were observed; meanwhile, in other research, average CO emissions did not statistically differ between 3% biochar and control piles [62]. Hellebrand and Schade demonstrated that the CO produced during composting is generated by abiotic reactions and not by microorganisms [64], but our own research shows that CO emissions differ in biotic and abiotic conditions [63], and there are a lot of different pathways when CO can be generated during the composting by microorganisms [65]. As biochar can positively affect the microbial community—the population of actinomycetes, cellulolytic and proteolytic bacteria showed increases over the period of composting [66]—this effect should be examined in future studies.

##### H_2_S Emissions

H_2_S has the highest odor potential generated during the composting process and is produced by the decomposition of sulfur-containing organic components by sulfate-reducing bacteria under anaerobic conditions. Most biochars are alkaline [67], and have a relatively high catalytic center dispersion in the pore system, thus, making them conducive for the effective oxidation of H_2_S. The emissions decreased with temperature, which was consistent with previous studies, when H_2_S was released mainly in the thermophilic phase (Appendix A). The H_2_S emission rate increased rapidly in all treatment groups, which was mainly attributed to the reduction in O_2_ content in the piles due to the rapid decomposition of OM by aerobic microorganisms. H_2_S was generated from two main pathways: the anaerobic decomposition of proteins and other sulfur-containing compounds coupled with the anoxic synthesis of sulfate-reducing bacteria [68]. The observed cumulative emissions were similar to those observed before by Ouyang, when the maximum emissions of all treatments were 200 µg∙g^–1^ d.m. [69], during sludge aerobic composting. In this study, highest emissions were observed with the addition of B650, >200 µg∙g^–1^ d.m., and the lowest with the B550 variant, ~50 H_2_S µg∙g^–1^ d.m. at a temperature of 50 °C and 100 H_2_S µg∙g^–1^ d.m. at 60 °C and 70 °C. Ouyang et. el. observed that the addition of biochar reduced from 12.91% to 50.47% of H_2_S. The biggest H_2_S reduction was observed at 50 °C with the addition of B550, but there was no direct effect of the biochar dose. Observed emissions mostly increased with temperature, and the effect of compost biochars became unclear—both the reduction and increases in H_2_S emissions with the addition of biochar were observed (Figure 4). Ky Nguyen et al. suggested that biochar could reduce the H_2_S emission, increasing with a proportion of biochar proportion of up to 20% [70]. On the other hand, ventilation of the piles and enhancing the microbial activity, along with effective reduction in the local anaerobic zone in the piles can reduce the sources of H_2_S production. Awasthi et al. showed that the pore diameter of biochar in the range of 25~45 μm was favorable for H_2_S adsorption [71]. According to the BET results, the pore diameter of compost biochars was 2.2, 1.6, and 1.4 nm for B550, B600, and B650, respectively, which is below the favorable level for H_2_S mitigation.

##### NH_3_ Emissions

Biochars influence the productivity of the composting process by enhancing microbial performance, altering physicochemical properties, enhancing substrate degradation, and resulting in rapid humification and gas emissions [72,73]. Biochars can be also used for effectively reducing ammonia emissions during composting, but the biochar properties affect NH_3_/NH_4_^+^ adsorption capacity and microbial growth, which may result in reducing or promoting ammonia emissions in the composting process [34]. In this study, observed emissions were in the range of 0–200 µg∙g^–1^ d.m, with no specific pattern—no effects of biochar, dose, or temperature of composting were observed (Figure 4; Appendix A). Recorded emissions were much lower than those observed by He et al. who reported 400 µg∙g^–1^ d.m. (powdered biochar) to even 800 µg∙g^–1^ d.m. (granular biochar) in 10 days of pig manure/wheat straw composting and confirmed that powdered biochar is most suitable for controlling NH_3_ emissions [1]. Additionally, the addition of compost biochars’ usually has an effect on the significant increase in observed emissions. It was observed that low-temperature biochars (300 °C) mitigated NH_3_ emissions better than high-temperature biochars (700 °C), possibly due to higher CEC [34]. The temperature used for production of biochars in the range of 550–650 °C could be not optimal for effective NH_3_ mitigation. As an example, a high SSA is reportedly required for biochars to be effective composting amendments because they may enhance adsorption capacity and microbial activity [74]. High-temperature biochars (B600, B650) with higher SSAs reduced NH_3_ emissions during composting slightly better than B550 (Appendix A). In addition, the pores on the biochar surface can become partially blocked during composting, which can promote anaerobic conditions, as evidenced by the observed H_2_S emissions. Pore blockage can limit the potential of high SSA to mitigate emissions during composting as this interferes with the penetration of adsorbents and microorganisms into the composting matrix [75]. Furthermore, as the pyrolysis temperature increases, biochars become more aromatic, and toxicity increases (PAHs) [76]. These factors may decrease and change microbial activity (promote nitrification or denitrification) during composting, which has an effect on the observed greater emissions of ammonia. Other possible explanations of the low effectiveness of compost biochars for ammonia mitigation could be an increase in the content of nutrients and porosity which provide favorable conditions for microbial growth and hence accelerate the immobilization of ammonia into the microbial biomass [17].

#### 3.2.3. Effect of Biochars on Nutrients and Heavy Metal Content during Initial Stages of Composting

##### C, H, N, S Changes

The elemental composition of the compost was evaluated by determining the content of C, H, N, and S in the biochar-amended feedstock after 10 days of incubation. In the case of carbon content in the post-incubated feedstock (Figure 5), the highest amounts were observed for samples with the addition of biochars produced at the temperature of 650 °C. It is widely documented that the pyrolysis temperature is crucial for the carbonization rate of biochar. High-temperature biochars are characterized by a polymerized, aromatic structure [35,77], which results in increased carbon content and was confirmed in our study. Although the effect of the condition of pyrolysis on compost biochars’ C content is clear, the dose of biochar did not have a significant effect on the C content in feedstock among the tested treatments, since for most doses the differences in C content were insignificant, reaching very similar values (no significant statistical change was observed). The mixing of stable, well-carbonized biochar with easily decomposable compost has been suggested as a solution for the enhancement of process performance and prevention of carbon losses [78]. For example, Gao et al., who tested the effect of co-composting with 300–400 °C biochar, noted that the amendment significantly increased the C content of composted material, which rose to 32.0% from 24.8% [79]. In the research of Darby et al., the addition of biochar produced at 600 °C increased the C content of the co-composted material from 28.2% to 35.1% [38]. However, in our study, the significant effect of biochar on C content in composted feedstock was not confirmed, which may be an effect of the relatively short time of the experiment. Nevertheless, the results of C content stay in line with CO_2_ emissions, as the highest carbon content was noted in treatments with lowest carbon dioxide emissions—incubated at 70 °C and amended with B650. It is noteworthy that the C content measured in our experiments is similar to findings of the aforementioned authors, reaching 35–40%.

Carbon and nitrogen are known as sources of energy for microbes, determining microbial activity in composted material. In general, biochar amendment does not directly supply the nitrogen and its impact on the compost quality results from an effect on nitrogen dynamics [80]. Biochars have the ability to absorb nitrogen ions, thus reducing the losses and enhancing the content of the element in composted material. Nitrogen content in analyzed samples varied between 2 and 3%, with no statistically significant effect of the biochar dose. Despite no statistical differences, it can be noted that in the majority of treatments, N content in samples without biochar amendments was noticeably lower than in treatments with 3–12% of biochar. As the incubation temperature of the material increases during the composting process, the percentage of the element decreases. Lower values of N were also seen in the case of biochar produced at 650 °C. Although some authors claimed that biochar-amended piles showed significantly higher N content in comparison to control treatments, in our study we did not clearly confirm this trend. Nevertheless, processes that have an impact on nitrogen dynamics are complex, and reports of enriching N content in the biochar-composted material involve longer incubation times, of up to 40 days [81,82]. On the other hand, presented observations support the theories that despite the role of biochar in the mitigation of carbon and nitrogen losses from composting processes, the final C/N ratio in the material is not significantly affected by the addition of biochar.

In the case of H, the content of the element in the analyzed material did not exceed a value of 5% (Appendix A Appendix A). The highest H values were observed in material composted with biochar produced at 650 °C and incubated at 60 °C, while the same biochar incubated at 70 °C showed the most stabilized values for all doses. From a statistical point of view, no correlation was observed between the parameters presented and the H content of the samples. For the S content, the greatest discrepancies were seen especially between the temperature of the pyrolysis process and the content of the element in the samples. It was observed that as the temperature of the pyrolysis process increased, the percentage of S decreased (Appendix A). Increasing the pyrolysis temperature causes the loss of volatile, easily decomposable elements such as H and S; therefore, their content in biochar amendment was low and enrichment of the composted material was not expected [83]. The greatest differences between the doses of biochar in the material were shown in samples where biochar produced at 550 °C was used. The addition of biochar obtained by pyrolysis at 550 °C caused a decrease in the proportion of S, especially for the higher doses. As sulfur and hydrogen are not crucial nutrients for evaluating properties of the organic material, the literature lacks reports on the effect of biochar on H and S content during composting. Hagemann, who evaluated the impact of a 4.3% dose of three different biochars on manure composting, found a mixed effect on sulfur content depending on the biochar type, whereas the H content was not elaborated [72]. 

##### Nutrients and Heavy Metal Changes

The compositional makeup of organic matter that has undergone the process of composting can be analyzed to deduce its fertilizing potential and ascertain its suitability for application in soil. This evaluation takes into consideration the presence of both elemental, trace, and heavy metal components. This study elaborates on the total amounts of biologically significant elements like Ca, Mg, K, Na, P, and trace metals including Zn, Pb, Cd, Cu, Cr, Ni, and Hg (Figure 5). The presence of specific elements within a material can exert a significant influence on the properties of the soil into which it is administered. This influence may be advantageous or detrimental, depending on the nature of the elements present and the intended outcome of the application. Therefore, it is imperative to carefully evaluate the composition of any material prior to its use in soil management. Parameters such as the incubation temperature of the material, the temperature of biochar production, and the dose of biochar added to the feedstock were checked.

The addition of biochar to composted feedstock altered the total content of macronutrients: Ca, Na, Mg, and K. Considering the temperature of pyrolysis, the presence of nutrients in biochar-amended compost was the highest in the B550 treatments and decreased along with the increasing temperature of pyrolysis (Figure 6). The literature reports claim that a higher temperature of pyrolysis promotes the concentration of nutrients in biochar; however, our study did not confirm this trend [84]. The Ca, Na, Mg, and K content in tested biochars was higher than in compost before pyrolysis; however, B650 showed the lowest nutrient concentration, which was reflected in the compost–biochar composition. The variability in the content of nutrients is an effect of their different volatility and stability, affected by pyrolysis conditions [85]. Considering the biochar dose, some significant effects on composted feedstock were observed. In every treatment, the concentration of Na and Mg significantly increased, suggesting that biochars introduce additional elements and also prevent their losses in the composting process. For phosphorus and potassium, the effect was not that clear, as some significant decreases were observed in comparison to the unamended feedstock. For example, 3 and 9% of biochar reduced the content of phosphorus, similar to the observations of Hagemann et al., who explained this phenomenon by P losses during the composting process [86]. Significantly reduced amounts of total K in co-composted biochar is in line with the findings of Mujtaba et al. who examined fruit waste and vegetables as a feedstock [87]. In general, higher biochar doses in the co-composting process yield a material with elevated content of macronutrients, as an additional source of the elements is introduced, with the exception of K.

Total trace metal content in organic amendments is one of the most important factors determining their suitability for application purposes. The results of selected trace metal contents in co-composted materials were much more varied than for other parameters studied. In general, considering the biochar type, at higher temperatures of pyrolysis the concentration of Cd, Cu, and Hg decreased; meanwhile, the content of Ni, Cr and Pb tended to be the highest in treatment with B600 (Figure 6). For Hg, no significant differences were observed between tested pyrolysis conditions. The results suggest that during pyrolysis, heavy metals were enriched in the biochars, as an effect of the better thermo-stability of metallic elements than other organic compounds present in the pyrolyzed biomass [88]. The fact that noted enrichment levels vary with tested metals may be an effect of metal speciation, influencing the temperature of decomposition [89].

Changes in metal concentrations during the composting process statistically differed between incubation temperatures. For all tested heavy metals, their mean content was the lowest after 10 days of co-composting at 70 °C, and this difference was statistically significant for Ni, Cr, and Cu. Fluctuations in trace metal content throughout the composting were observed by other authors [90]; however, the direction of the changes was often varied, depending on metal type. Composting and maturation processes decrease the heavy metal content and availability, and at higher temperatures, this effect was more significant [91].

Considering biochar doses on the content of potentially harmful metals in composted feedstock, it can be seen that a higher amount of introduced biochar amendment often resulted in an elevated metal content. This trend was clear for the 12% and 15% dose—a large amount of biochars, enriched in metals during pyrolysis process, led to enhanced concentrations in co-composted substrate. In the majority of cases, that enhancement was statistically significant, with the exception of Hg. Hagemann et al. who examined compost–biochar (with a dose of 4.3%), noted that the amendment did not significantly elevate the content of trace metals [84], and this is in agreement with our observations, where small doses (3–6%) of biochar were applied. Therefore, although small doses of biochars often do not have a statistically significant effect on HM content, higher doses can result in elevated metal concentrations, because of the additional pool of elements introduced with enriched, pyrolyzed materials [92]. This should be the subject of special attention when using compost–biochar in practice, due to the possibility of introducing harmful metals into the soil, their uptake by plants, and their inclusion in the food chain [93].

## 4. Conclusions

This study shows the effect of the addition of compost biochars to the composting process—the addition of compost biochars effectively decreases the CO_2_ emissions and increases the nutrient content in the composting matrix. The effective use of compost biochars for gas mitigation during the intensive phase of composting demands the specific conditions of the process. It is presumed that the main mechanism of CO_2_ mitigation was the addition of biochar with its inorganic content with a high composition of nutrients that enhance the microorganisms’ activity. This effect should be examined in future studies. The following recommendations in compost biochars’ application in compost can be formed:The use of compost biochars was the most effective in CO_2_ mitigation—total reduction in emissions > 80% at 60 °C; lower reduction < 20% at 50 and 70 °C.The effect for CO, H_2_S, and NH_3_ mitigation was unclear; surprisingly, the reduction in CO emissions was observed at a temperature of 50 °C, with 30–60% effectiveness for B550, and 20–30% for B600; however, at 70 °C, an increase of >300% in CO emissions was observed; for H_2_S, the most effective was incubation at 50 °C–a reduction > 50%, with no positive effect for 60 and 70 °C.No positive effect for NH_3_ mitigation was observed, probably due to the high pH content of biochar.The addition of small doses (3–6% d.m.) of compost biochars reduced the observed emissions and significantly improved (the best treatment was B650, incubation at 60 °C, with 3% of the addition of biochar) the content of nutrients in composting matrix (P, K, Mg, Ca). In the same treatments, the content of trace metals was safe for the future use of compost in agriculture.

This study confirms that the use of pyrolysis for compost materials improves the properties of compost biochars, and has good potential for enhancing the composting process. This gives a new alternative niche for managing low quality compost and to become a part of the bioeconomic approach.

## Figures and Tables

**Figure 1 materials-17-00563-f001:**
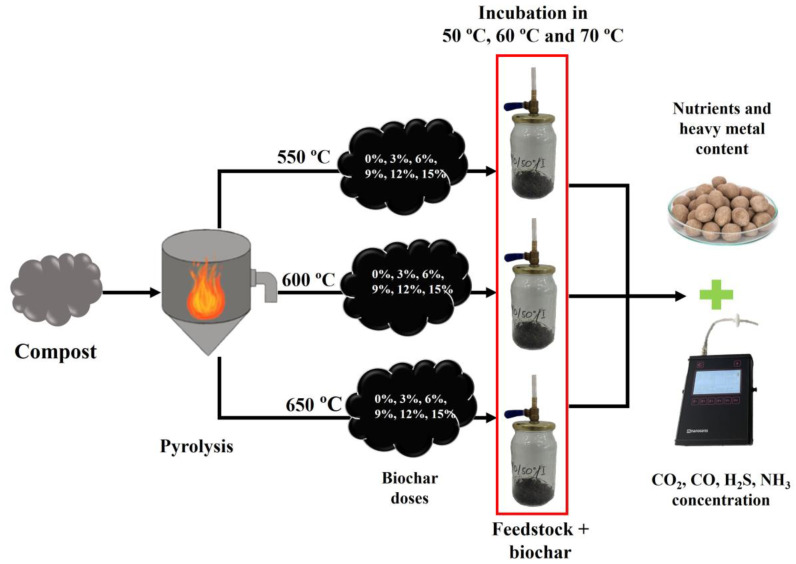
Experiment configuration.

**Figure 2 materials-17-00563-f002:**
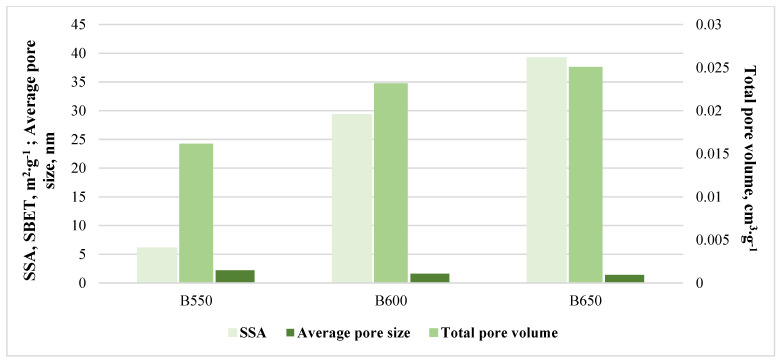
BET results for compost biochars.

**Figure 3 materials-17-00563-f003:**
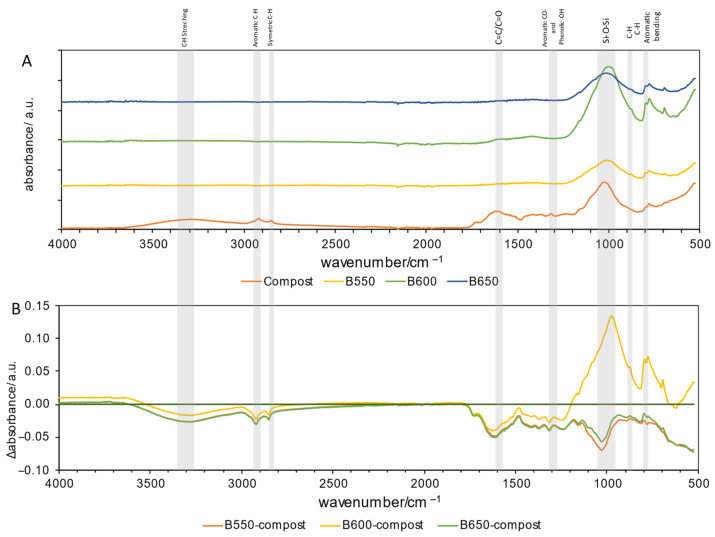
FTIR results of compost and compost biochars; (**A**) shows the FTIR spectra of compost and compost biochars (B550, B600, B650), and (**B**) shows the subtraction spectra of samples. The line indicates zero absorbance change for each subtraction spectrum.

**Figure 4 materials-17-00563-f004:**
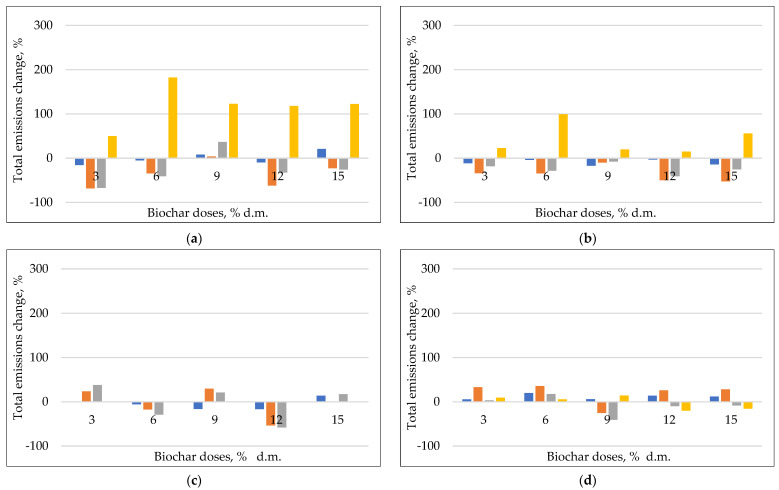
Total gas emission change, during the first 10 days of composting process at laboratory scale, (**a**) B550, incubation at 50 °C, (**b**) B600, incubation at 50 °C, (**c**) B650, incubation at 50 °C, (**d**) B550, incubation at 60 °C, (**e**) B600, incubation at 60 °C, (**f**) B650, incubation at 60 °C, (**g**) B550, incubation at 70 °C, (**h**) B600, incubation at 70 °C, (**i**) B650, incubation at 70 °C.

**Figure 5 materials-17-00563-f005:**
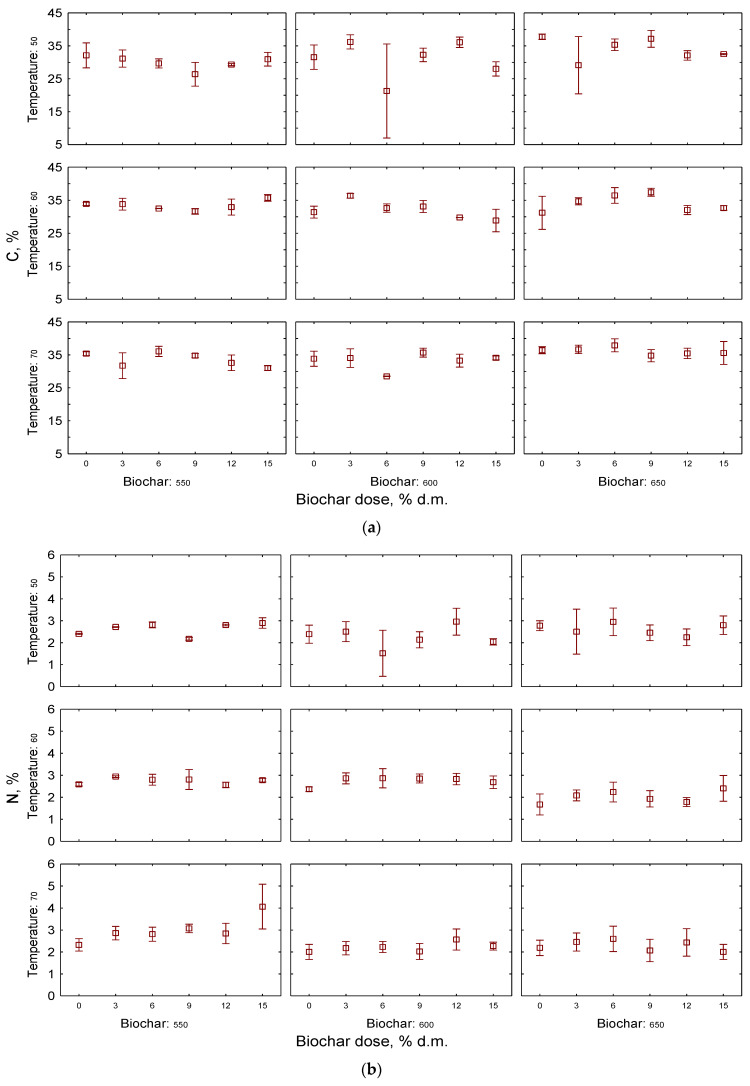
Effect of biochar dose for (**a**) C, % and (**b**) N, %, during the first 10 days of composting process at a laboratory scale; no statistical differences were observed.

**Figure 6 materials-17-00563-f006:**
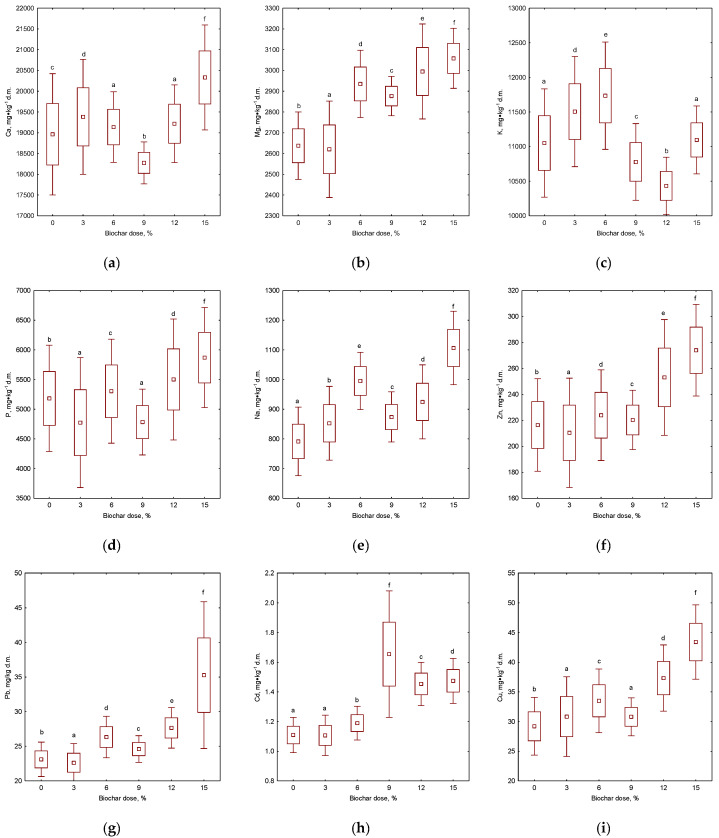
Effect of compost biochars’ addition dose for nutrients and heavy metal content in substrate after 10 days of composting (**a**) calcium (Ca), (**b**) magnesium (Mg), (**c**) potassium (K), (**d**) phosphorus (P), (**e**) sodium (Na), (**f**) zinc (Zn), (**g**) lead (Pb), (**h**) cadmium (Cd), (**i**) copper (Cu), (**j**) chrome (Cr), (**k**) nickel (Ni), (**l**) manganese (Mn), and (**m**) mercury (Hg). Letters (a, b, c, d, e, f) indicate the homogeneity group according to Tukey’s test at significance level *p* < 0.05.

**Table 1 materials-17-00563-t001:** Characteristics of compost, compost biochars’ variants, and feedstock (mix of green waste and sewage sludge).

Parameters	Compost	Feedstock	Biochars’ Variant		
			B550	B600	B650
pH	7.32 ± 0.04	4.77 ± 0.04	8.61 ± 0.03	9.10 ± 0.03	9.50 ± 0.04
LOI, % d.m.	24.80 ± 0.00	82.01 ± 3.70	11.0 ± 0.00	11.9 ± 0.00	9.30 ± 0.00
Ca, mg∙kg^−1^	17,760 ± 109	14,878 ± 92.5	19,795 ± 175	18,460 ± 105	15,975 ± 80
K, mg∙kg^−1^	6155 ± 65	8608 ± 30	7865 ± 65	7665 ± 60	7390 ± 65
Mg mg∙kg^−1^	2697 ± 29	2405 ± 22	3417 ± 35	3382 ± 35	2967 ± 10
Na, mg∙kg^−1^	1793 ± 1.1	773 ± 5.8	2263 ± 1.5	2168 ± 1.0	2143 ± 1.0
P, mg∙kg^−1^	5835 ± 1167	6300 ± 1260	6865 ± 1373	6953 ± 1391	6749 ± 1350
Pb, mg∙kg^−1^	35.52 ± 7.11	25.57 ± 0.70	45.81 ± 9.16	39.59 ± 7.92	31.01 ± 6.20
Cd, mg∙kg^−1^	1.66 ± 0.33	0.800 ± 0.00	1.98 ± 0.40	1.91 ± 0.38	1.15 ± 0.23
Zn, mg∙kg^−1^	553.8 ± 110.8	253.3 ± 1.5	585.3 ± 117.3	511.8 ± 102.3	426.8 ± 85.4
Cr, mg∙kg^−1^	60.7 ± 12.2	33.0 ± 1.6	141.3 ± 28.3	96.3 ± 19.3	56.6 ± 11.3
Ni, mg∙kg^−1^	11.9 ± 2.4	20.7 ± 0.05	43.2 ± 8.6	32.1 ± 6.4	10.6 ± 2.1
Cu, mg∙kg^−1^	63.7 ± 12	44.03 ± 0.2	72.7 ± 13.8	62.1 ± 11.7	45.2 ± 8.3
Mn, mg∙kg^−1^	362.3 ± 72.5	253.3 ± 1.5	416.8 ± 83.4	391.5 ± 78.3	341 ± 68.2
Hg, mg∙kg^−1^	0.106 ± 0.03	0.08 ± 0.02	<0.001	<0.001	<0.001
C, %	13 ± 3	35.89 ± 4.1	8.3 ± 1.7	9.3 ± 1.8	10 ± 3
H, %	1.2 ± 0.2	4.82 ± 0.9	0.21 ± 0.04	0.23 ± 0.05	0.05 ± 0.2
N, %	1.1 ± 0.2	2.37 ± 0.2	0.48 ± 0.1	0.62 ± 0.12	0.44 ± 0.2
S, %	0.38 ± 0.08	1.64 ± 0.05	0.31 ± 0.06	0.33 ± 0.07	0.31 ± 0.08
O, %	8.7 ± 0.4	37.29 ± 1.9	1.7 ± 0.3	0.82 ± 0.2	0.4 ± 0.4

## Data Availability

The data presented in this study are available in the Appendix A.

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
