# Peer review of "Effective Carbon Dioxide Mitigation and Improvement of Compost Nutrients with the Use of Composts’ Biochar"

_materials, 2024, doi:10.3390/ma17030563_

Round 1

Reviewer 1 Report

Comments and Suggestions for Authors

This study was conducted to investigate composts’ biochar capabilities to reduce greenhouse gas emissions (GHG) including CO2 mitigation and improve the quality of compost by increasing the concentration of nutrients in the composting matrix.

The paper is well-structured, outlining the technique, and showcasing the findings and appropriate results, however, there are some comments:

The English of the paper is readable; however, I would suggest the authors to have it checked, preferably by a native English-speaking person, to avoid any mistakes.

Abstract:

lines 15-17: Please rewrite.

Line 22: The authors stated that the emissions were measured every day. For how long or how many times???

At the end of the abstract please add the conclusion of your research in one or two sentences.

The authors stated that even small doses of compost's biochar (3–6% d.m.) can reduce total measured gas emissions. Please add the best treatment, and which level had the highest impact on reducing gas emissions.

Introduction:

Lines 32-35: add reference(s).

It would be great if could bring some information about biochar-compost on crop yield and soil properties in the second paragraph. In the second paragraph, you stated the effect of biochar on gas emissions, soil fertility, and … It's a question for the reader what is the effect of biochar compost on crop yield? So, I would suggest that in two or three sentences explain the advantages of biochar compost on crop yield. Here is a published work that fits with your scops, and you can use it to improve this gap. http://dx.doi.org/10.1007/s10333-022-00912-8

The innovation of the study should be clear at the end of this section. The background of the research is not enough. The authors mentioned that …. This study aims for the first time to test the composts’ biochar capabilities to reduce greenhouse gas emissions (GHG) …., but there are some literatures near to the current research, so the authors should compare the current research with previous ones and bold the gap of knowledge of compost biochar. Please add some previous research here and improve the introduction.

Material and Methods:

Please add reference(s) for biochar characteristics analyses (lines 101-107).

Line 127: then2, ….??????

Results and discussion

Lines 220-221: Unclear. Please rewrite.

Table 1: standard deviation (SD) for some parameters is missing. Please check and correct the table.

Line 231: for the current research, the authors use the FTIR spectroscopy, please add the FTIR details in the material and method section.

Figure 4: The resolution of the figures is very low and unclear. Please bring the figures in another way.

Line 302: 3.2.2.1. CO2 emissions

The discussion of this section is not appropriate, since carbon dioxide is the main gas generated during the bio-oxidative phase of composting, I would suggest that to expand the discussion with more comparisons.

 Line 330: 3.2.2.2. CO2 emissions à CO emissions

Conclusion:

This section is well written.

Comments on the Quality of English Language

Minor editing of English language required. The English of the paper is readable; however, I would suggest the authors to have it checked, preferably by a native English-speaking person, to avoid any mistakes.

Author Response

Dear Reviewer,

Thank you for all of your comments. Our responses are enclosed below.

Reviewer 2 Report

Comments and Suggestions for Authors

In this manuscript (materials-2789630) entitled "Effective CO2 mitigation and improvement of compost nutrients with the use of composts’ biochar" submitted to Materials, Sylwia Stegenta-DÄ…browska and colleagues have analyzed the gaseous emissions during the first 10 days of the composting process using biochars obtained from mature composts. Authors demonstrated that even small doses of compost's biochar can reduce up to 50% of the total measured gas emissions while also increasing the nutrient content of P, K, Mg, and Ca significantly. This research is interesting and convincing, but minor points need to be addressed to improve the quality of this manuscript.

1. For Figure 1, biochors characteristics should be analyzed with microscopy and displayed in the revised Figure.

2. For Figure 2 and Table 1, analysis in significance of difference should be performed. Please exhibit the significance of difference in the revised Figure and Table.

3. For Figure 4, Total gas emissions change at 0 (zero) % d. m biochar should be analyzed and shown in the revision.

4. For Figure 5, Authors should consider to split this Figure into two panels to show the C% and N% separately.

Author Response

Dear Reviewer,

Thank you very much for the corrections to our manuscript. Our responses are enclosed below.

Round 2

Reviewer 2 Report

Comments and Suggestions for Authors

Authors have addressed my concerns in the revision.